# Mask adherence and rate of COVID-19 across the United States

**Charlie B. Fischer°, Nedghie Adrien (iD)°, Jeremiah J. Silguero, Julianne J. Hopper, Abir I. Chowdhury, Martha M. Werler (iD) ***

Boston University School of Public Health, Boston, MA, United States of America

° These authors contributed equally to this work.
* werler@bu.edu

## Abstract

Mask wearing has been advocated by public health officials as a way to reduce the spread of COVID-19. In the United States, policies on mask wearing have varied from state to state over the course of the pandemic. Even as more and more states encourage or even mandate mask wearing, many citizens still resist the notion. Our research examines mask wearing policy and adherence in association with COVID-19 case rates. We used state-level data on mask wearing policy for the general public and on proportion of residents who stated they always wear masks in public. For all 50 states and the District of Columbia (DC), these data were abstracted by month for April — September 2020 to measure their impact on COVID-19 rates in the subsequent month (May — October 2020). Monthly COVID-19 case rates (number of cases per capita over two weeks) >200 per 100,000 residents were considered high. Fourteen of the 15 states with no mask wearing policy for the general public through September reported a high COVID-19 rate. Of the 8 states with at least 75% mask adherence, none reported a high COVID-19 rate. States with the lowest levels of mask adherence were most likely to have high COVID-19 rates in the subsequent month, independent of mask policy or demographic factors. Mean COVID-19 rates for states with at least 75% mask adherence in the preceding month was 109.26 per 100,000 compared to 249.99 per 100,000 for those with less adherence. Our analysis suggests high adherence to mask wearing could be a key factor in reducing the spread of COVID-19. This association between high mask adherence and reduced COVID-19 rates should influence policy makers and public health officials to focus on ways to improve mask adherence across the population in order to mitigate the spread of COVID-19.

## Introduction

The global pandemic of SARS-CoV-2 has overwhelmed health care systems, marked by peak numbers of hospital and intensive care unit admissions and deaths [1,2]. Mask wearing has been advocated by public health officials as a way to reduce the spread of COVID-19 [3–5]. In the United States, policies on mask wearing have varied from state to state over the course of the pandemic [6]. For the period of April 1 through October 31, 2020, less than half of states

**Data Availability Statement:** Data are available in the paper and its Supporting Information file.

**Funding:** The authors received no specific funding for this work.

**Competing interests:** The authors have declared that no competing interests exist.

had issued a mandate for mask wearing in public and nearly a third had not made any recommendation. Even as more and more states encourage mask wearing [7], many citizens still resist the notion [8]. Individuals' mask wearing behaviors are not only influenced by recommendations and mandates issued by state leaders, but also by print, televised, and social media [9]. Thus, adherence to mask wearing in public remains a challenge for mitigating the spread of COVID-19.

Public health policy-making requires navigating the balance of public good and individual rights [10]. The adoption of universal masking policies is increasingly polarized and politicized, demanding that public health authorities balance the values of health and individual liberty. Adherence to public policy is influenced by a complex interplay of factors such as public opinion, cultural practices, individual perceptions and behaviors [11], which are difficult to quantify. The politicization of COVID-19 epidemiology [9,12] has further complicated policy-making, messaging, and uptake. Nevertheless, adherence is essential for policy effectiveness. Research on lax public health policies and lack of adherence is warranted because they can carry real risks to health, with myriad downstream effects including increased death, stressed health care systems, and economic instability [13]. We examined the impact of state-based mask wearing policy and adherence on COVID-19 case rates during the summer and early fall of 2020 in order to quantify this effect.

## Methods

For all 50 states and D.C., data on mask wearing and physical distance policies, mask adherence, COVID-19 cases, and demographics were abstracted from publicly available sources. We utilized the COVID-19 US State Policy Database, created by Dr. Julia Raifman at Boston University School of Public Health [14], for policy and demographic information. We abstracted data on whether the state issued a mandate of mask use by all individuals in public spaces, and if so, the dates of implementation and whether the mandate was enforced by fines or criminal charge/citation(s). For policies on physical distancing, we recorded whether a stay-at-home order was issued and, if so, when. For mask adherence levels, we utilized the Institute of Health Metrics and Evaluation (IHME) COVID-19 Projections online database [15], which holds data collected by Facebook Global in partnership with the University of Maryland Social Data Science Center [16]. We abstracted daily percentages of the population who say they always wear a mask in public. To calculate monthly COVID-19 case rates, we abstracted the number of new cases reported by the U.S. Centers for Disease Control and Prevention (CDC) [17] and state population sizes in 2019 [18].

### Mask wearing policy

We categorized the existence of a mask policy as "None" if there was no requirement for face coverings in public spaces, "Recommended" if required in all public spaces without consequences, and "Strict" if required in all public spaces with consequences in the form of fine(s) or citation(s). We combined the Recommended and Strict groups into "Any" policy. States and D.C. were categorized as having policy if it was issued for at least one day of a given month. Although Hawaii's governor did not issue a mask wearing policy until after October 2020, we considered that state to have a policy because mayors of the four populous counties had mandated mask wearing earlier in the pandemic.

### Mask wearing adherence

We calculated the average mask use percentage by month for April–September, 2020. For each month, the distributions of mask adherence across all 50 states and D.C. were categorized into

quartiles, meaning the cut-off values for each quartile may be different from one month to another. Mask adherence was classified as low if in the lowest quartile and as high if in the highest quartile. We also identified states with average mask adherence ≥75% in a given month.

### COVID-19 rates

We calculated the number of new cases in each month, for each state and D.C. Rates were the number of new cases divided by the population in 2019. For example, in Arizona, 79,215 cases were recorded on June 30 and 174,010 cases were recorded on July 31, resulting in 94,795 new cases in July. We divided the monthly number by 2.2 to obtain the number in a two-week period (43,088). The 2-week rate in July in Arizona = 43,088 cases/7,278,717 population in 2019 = 0.00592 or 592 per 100,000. We classified a state and D.C. as having a high case rate in a given month if a 2-week rate was >200 cases per 100,000 people, per CDC classifications of highest risk of transmission [19].

### Covariates

Based on CDC at-risk guidelines for COVID-19 [20], we considered non-Hispanic Black, Hispanic, age, and population density as potential confounders. Data on population distributions from the COVID-19 US State Policy Database [13] came from the US Census. For demographic data, we dichotomized population proportions at whole values that approximated the highest quartile of the distributions. Specifically, we created the following categories: >15% non-Hispanic Black, >15% Hispanic, median age >40 years, and population density >200 people per square mile, which corresponded to 74.5%, 78.4%, 82.4%, and 78.4% of the distributions, respectively. Policy data on physical distancing were dichotomized as any versus no stay-at-home order during the April 1 to October 31, 2020 interval.

### Statistical analysis

Our analyses took into consideration the delayed effect of mask wearing and policies on COVID-19 health outcomes. Thus, policy and adherence levels in a given month were contrasted with lagged COVID-19 case rates in the subsequent month. Both mask policy and mask adherence for states and D.C. were cross-tabulated with high case rates in the subsequent month. Logistic regression models were used to estimate the odds ratio and 95% confidence intervals for high case rates in the subsequent month associated with average mask adherence (as a continuous variable). Models were unadjusted, adjusted for no mask policy (Model 1), and adjusted for no mask policy in previous month, no stay-home order, >15% population non-Hispanic Black, >15% population Hispanic, median age >40 years, population density > 200/square mile (Model 2).

## Results and discussion

States in COVID-19 high-risk categories are listed in Table 1. Because stay-at-home order, mask wearing policy, mask adherence, and COVID-19 rates can vary from month to month, we listed those states with consistent classifications across the period April through September (or May through October for COVID-19 rates). Eleven states had no stay-at-home order, 15 had no mask policy, and four states had low adherence throughout this six-month period.

The list of states with high COVID-19 rates by month shows the initial wave in northeastern states in May, followed by a wave in southern states, and then spreading across the U.S. over the next four months (Table 2). Of the 15 states with no mask policy from April through

**Table 1. States with high COVID-19 population risk characteristics.**

| High risk category | States |
|---|---|
| >15% non-Hispanic Black | AL, AR, DC, DE, FL, GA, HI, LA, MD, MS, NC, SC, TN, VA |
| >15% Hispanic | AZ, CA, CO, CT, FL, IL, NJ, NM, NV, NY, RI, TX |
| Median age >40 years | CT, DE, FL, ME, MT, NH, NJ, PA, RI, VT, WV |
| Pop. density > 200/mile$^2$ | CA, CT, DC, DE, FL, HI, MA, MD, NH, NY, OH, PA, RI |
| No stay at home order | AR, CT, IA, KY, ND, NE, OK, SD, TX, UT, WY |
| No mask policy Apr-Sep | AZ, FL, GA, IA, ID, MO, MT, ND, NE, NH, OK, SC, SD, TN, WY |
| <25%ile mask adherence Apr-Sep | IA, KS, ND, SD |

September, 14 reported high COVID-19 rates in at least one month from May to October. Because high COVID rates were reported by only eight states in May and four states in June, we did not examine mask adherence or policy in the preceding April or May. Thus, subsequent comparisons of states with high COVID-19 rates by month focused on July, August, September and October. Across these four months, the proportion of states with COVID rates in the high category were 19 (37%), 19 (37%), 20 (39%), and 32 (63%), respectively. Eight states were reported to have at least 75% mask adherence in any month between June and September (AZ, CT, HI, MA, NY, RI, VT, VA); none reported a high COVID-19 rate in the subsequent month.

For mask adherence, the cut-off values for the low and high quartiles were 31% and 46% in June, 53% and 72% in July, 55% and 71% in August, and 55% and 68% in September. The proportions of states with high COVID-19 rates are shown for those in the low and high quartiles of mask adherence in the preceding month (Fig 1). Most states in the low quartile had high COVID-19 rates in the subsequent month. Indeed all 13 states in the low mask adherence group in September had high COVID-19 rates in October. In contrast, just one state in July, August, and September and three in October in the high quartile had high COVID-19 rates in the subsequent month. When we looked at states with ≥75% mask adherence (Arizona, Connecticut, Hawaii, Massachusetts, Michigan, New York, Rhode Island, Vermont), we found none had experienced a high COVID-19 rate in the subsequent month. Mean COVID-19 rates for states with ≥75% mask adherence in the preceding month was 109.26 per 100,000 compared to 249.99 per 100,000 for those with less adherence.

The proportions of states and D.C. with high COVID-19 rates were greatest for those with no mask wearing policy for the general public in the preceding month (Fig 2). Among states and D.C. with no mask wearing policy, 50 to 73% had high COVID-19 rates in the subsequent month. In contrast, 25% or fewer states with a mask wearing policy had high COVID-19 rates,

**Table 2. States with high COVID-19 rates.**

| COVID-19 >200 cases /100,000 | States |
|---|---|
| May | DC, DE, IL, MA, MD, NE, NJ, RI |
| June | AR, AZ, FL, SC |
| July | AL, AR, AZ, CA, FL, GA, ID, IA, KS, LA, MO, MS, NC, NV, OK, SC, TN, TX, UT |
| August | AL, AR, CA, FL, GA, ID, IA, IL, KS, LA, MO, MS, ND, NV, OK, SC, SD, TN, TX |
| September | AL, AR, GA, ID, IA, IL, KS, KY, MO, MS, MT, NE, ND, OK, SC, SD, TN, TX, UT, WI |
| October | AL, AK, AR, CO, DE, ID, IA, IL, IN, KS, KY, MI, MN, MO, MS, MT, NC, NE, ND, NM, NV, OH, OK, RI, SC, SD, TN, TX, UT, WI, WV, WY |
| Jul, Aug, Sep or Oct and no mask policy Jun–Sep | AZ, FL, GA, IA, ID, MO, MT, ND, NH, OK, SC, SD, TN, WY |

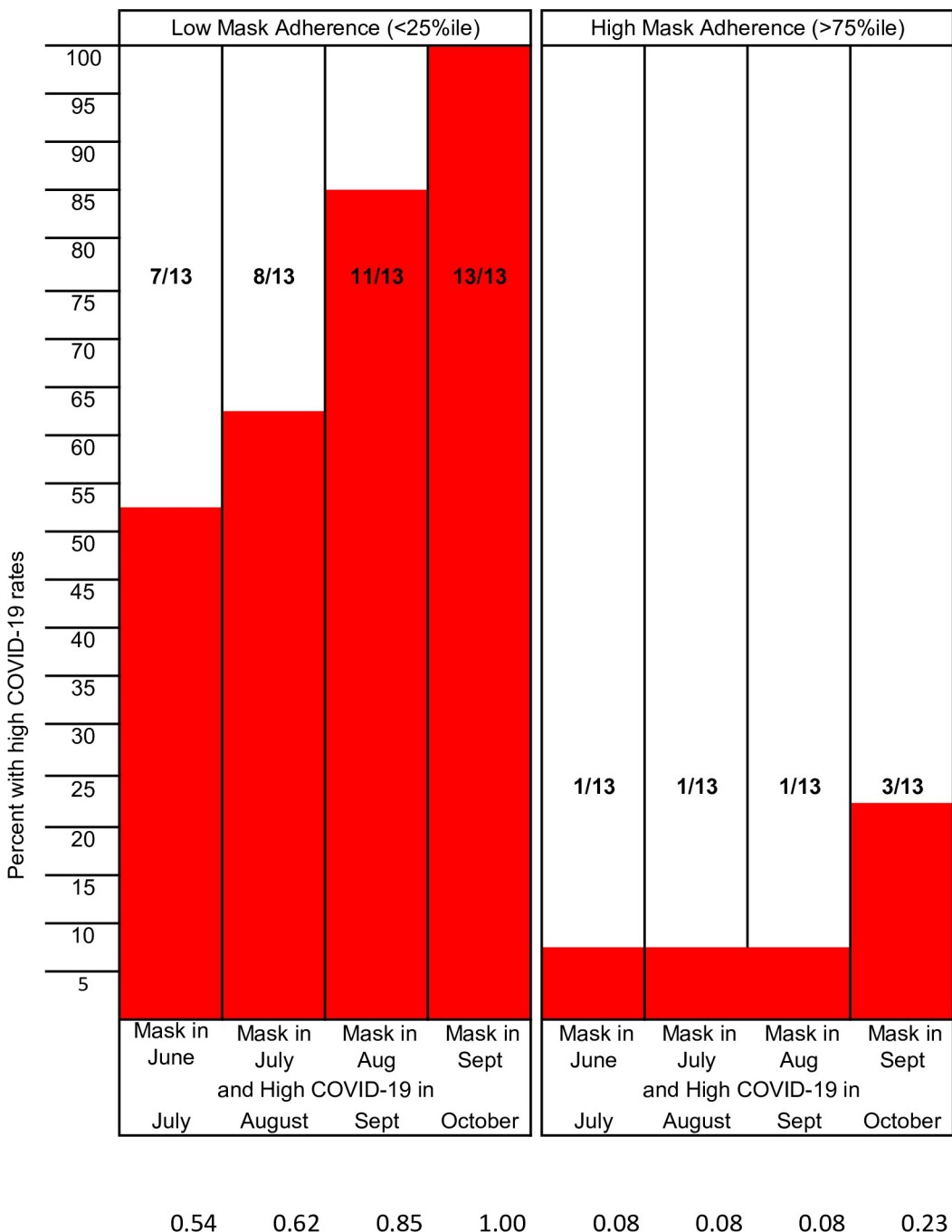

**Fig 1. Proportion of states with high COVID-19 rates among those in the low and high mask adherence quartiles in the preceding month.**

except in September when over half experienced high rates. Fourteen of the 15 states with no mask wearing policy for the general public for the entire four month period (June through September) reported a high COVID-19 rate. High COVID-rates were less frequent in states and D.C. with strict mask wearing policy than in states with recommended policy.

Looking more closely at October when COVID-19 rates increased across the US, we found average adherence was only 47% in September for the 11 states without a mask policy and high

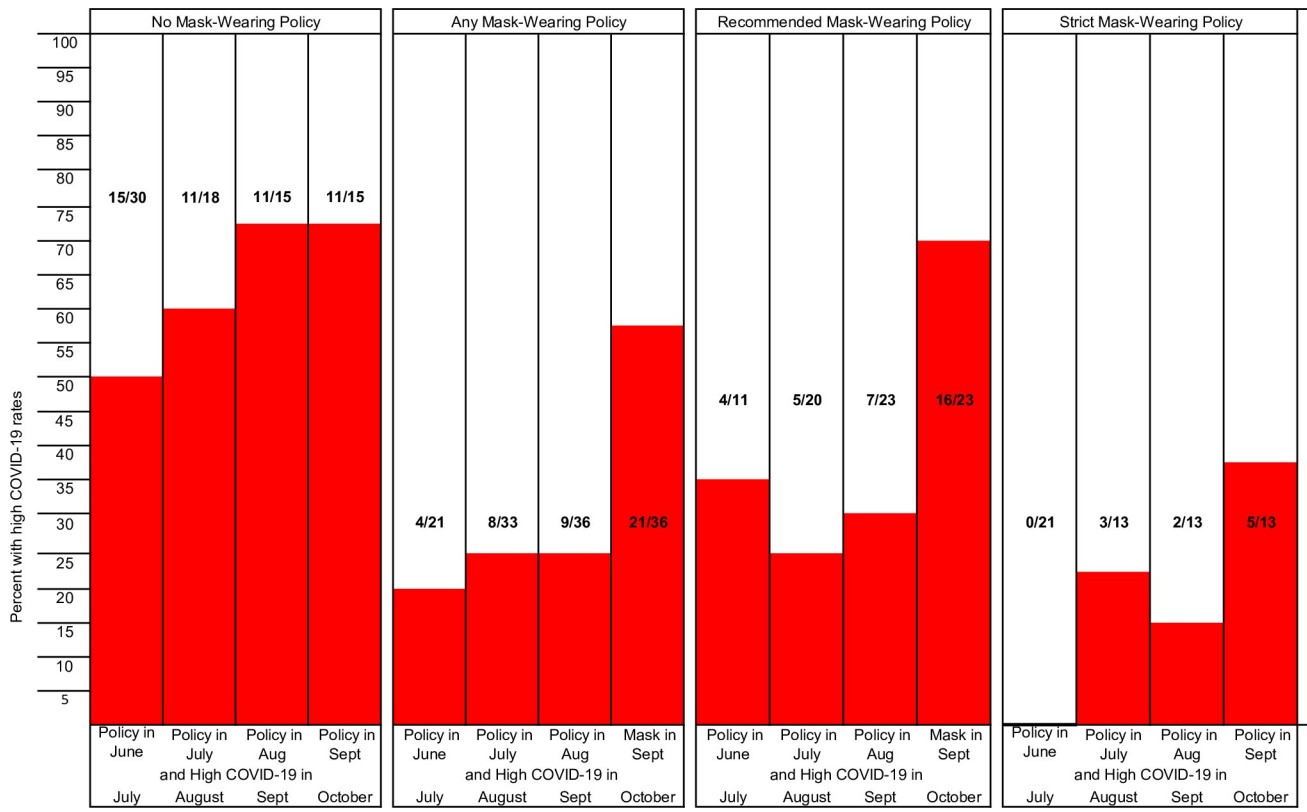

**Fig 2. Proportion of states with high COVID-19 rates among those no, any, strict, and recommended mask wearing policy in the preceding month.**

October COVID-19 rates. In contrast, average adherence was 68% in the 15 states with lower COVID-19 rates in October and any mask policy in September. Of note, there were no states with ≥75% in September.

Odds ratios and 95% confidence intervals for average mask adherence and mask policy for the general public are associated with high COVID-19 rates in the subsequent month (Table 3). Mask adherence was associated with lower odds of high COVID-19 rates, even after

**Table 3. State-level odds ratios and 95% confidence intervals (CI) for high versus lower COVID-19 rates in the subsequent month.**

| | | Unadjusted | | | Model 1[*] | | | Model 2[**] | | |
|---|---|---|---|---|---|---|---|---|---|---|
| | | OR | 95% CI | | OR | 95% CI | | OR | 95% CI | |
| June | Mask adherence, avg | 0.91 | 0.85, | 0.98 | 0.93 | 0.86, | 1.00 | 0.95 | 0.83, | 1.08 |
| | Any mask policy | 0.24 | 0.06, | 0.87 | 0.42 | 0.10, | 1.78 | 0.19 | 0.03, | 1.41 |
| July | Mask adherence, avg | 0.91 | 0.86, | 0.97 | 0.93 | 0.87, | 0.99 | 0.87 | 0.77, | 0.99 |
| | Any mask policy | 0.20 | 0.06, | 0.70 | 0.41 | 0.10, | 1.70 | 0.22 | 0.03, | 1.63 |
| August | Mask adherence, avg | 0.88 | 0.81, | 0.95 | 0.90 | 0.83, | 0.98 | 0.94 | 0.85, | 1.03 |
| | Any mask policy | 0.12 | 0.03, | 0.48 | 0.23 | 0.05, | 1.18 | 0.21 | 0.03, | 1.57 |
| September | Mask adherence, avg | 0.81 | 0.72, | 0.92 | 0.78 | 0.68, | 0.90 | 0.74 | 0.59, | 0.93 |
| | Any mask policy | 0.41 | 0.11, | 1.52 | 3.52 | 0.49, | 25.41 | 6.28 | 0.61, | 64.85 |

[*]Model 1, includes average mask adherence and any mask policy.

[**] Model 2, includes Model 1 and adjusted for no stay-home order, >15% population non-Hispanic Black, >15% population Hispanic, median age >40 years, population density > 200/mile2.

adjustment for mask policy and for demographic factors. For every 1% increase in average adherence in June, the fully adjusted odds ratios for high COVID-19 in July was 0.95, indicating a protective effect against high COVID-19 rates. Similar reductions in odds of high COVID-19 rates in August and September were observed for July and August mask adherence, respectively. The strongest association was for mask adherence in September; for every 1% increase in average adherence, the odds of a high COVID-19 case rate decreased by 26%.

Crude and adjusted odds ratios for any mask policy in relation to high COVID-19 rates in the subsequent month were below 1.0; but confidence intervals were wide. For mask policy and adherence in September in relation to high COVID-19 rates in October, collinearity caused the odds ratio to flip.

We were not able to measure statistical interactions between mask policy and adherence due to instability arising from small numbers. We did estimate odds ratios for mask adherence within subgroups of states with and without mask policy. Odds ratios indicating protection against high COVID-19 rates remained for all months and policy subgroups, ranging from 0.82 to 0.93 for states with any policy and from 0.60 to 0.95 for states with no policy.

## Interpretation

We show supporting evidence for reducing the spread of COVID-19 through mask wearing. This protective effect of mask wearing was evident across four months of the pandemic, even after adjusting the associations for mask policy, distance policy, and demographic factors. We observed some benefit of mask policy on COVID-19 rates, but the findings were unstable. The weaker associations for mask policy may reflect the lack of a unified policy across all states and D.C. and the inconsistent messaging by the media and government leaders. Indeed, issuing such a policy is not the same as successfully implementing it. Our observed associations should influence policy-makers and contribute to public health messaging by government officials and the media that mask wearing is a key component of COVID-19 mitigation.

Our observation that states with mask adherence by ≥75% of the population was associated with lower COVID-19 rates in the subsequent month suggests that states should strive to meet this threshold. The difference in mean COVID-19 rates between states with ≥75% and <75% mask adherence was 140 cases per 100,000. It is worth noting that no states achieved this level of mask adherence in September, which might account in part for the spike in COVID-19 rates in October. Of course, many other factors are could be at play, like the possibility of cooler weather driving non-adherent persons to indoor gatherings.

Our study accounted for temporality by staggering COVID-19 outcome data after adherence measures. Nevertheless, it is possible that average mask adherence in a given month does not capture the most effective time period that influences COVID-19 rates. For example, mask wearing in the two weeks before rates begin to rise might be a more sensitive way to measure the association. If this is true, we would expect associations between mask adherence and high COVID-19 rates to be even stronger. It is also possible that survey respondents misreported their mask wearing adherence; whether they would be more or less likely to over or under-report is open to speculation, but residents in states with mask wearing policy might over-report adherence to appear compliant. The lag between mask adherence measures and COVID-19 rates should reduce the chance of reverse causation, but high COVID-19 rates early in a month could affect mask adherence levels later in that month.

It is important to note that state level distributions of demographic factors do not account for concentrations or sparsity of populations within a given state. Further, our adjustment for demographic factors at the state population level may not represent the true underlying forces that put individuals at greater risk of contracting COVID-19. Though demographic factors

were measured as proportions of the population, even if they were considered to be indicators for individual level characteristics, they do not denote an inherent biologic association with the outcome and more likely reflect structural inequities that lead to higher rates of infection in minoritized populations. Another consideration is that access to COVID-19 testing appears to vary from state to state [21]. Our study was also limited by the lack of information on accessibility of COVID-19 testing; if less accessible testing is associated with less mask adherence, the associations we report here may be under-estimates.

Our analysis of state and D.C.-level data does not account for variations in policy, adherence, and demographic factors at smaller geographic levels, such as county-levels. Further analyses of more granular geographic regions would be a logical next step. Indeed, associations between mask policy, adherence and other factors may be obscured in states with many high density and low density areas.

## Conclusions

In conclusion, we show that mask wearing adherence, regardless of mask wearing policy, may curb the spread of COVID-19 infections. We recommend renewed efforts be employed to improve adherence to mask wearing.

## Supporting information

**S1 File.**
(CSV)

## Acknowledgments

We thank Dr. Julia Raifman Boston University School of Public Health for developing, maintaining, and providing open access to the COVID-19 US State Policy Database. We thank Drs. Eleanor J. Murray and Jennifer Weuve and the Boston University School of Public Health Epidemiology COVID-19 Response Corps for bringing together students and faculty for this project.

## Author Contributions

**Conceptualization:** Charlie B. Fischer, Jeremiah J. Silguero, Julianne J. Hopper, Abir I. Chowdhury, Martha M. Werler.

**Data curation:** Charlie B. Fischer, Jeremiah J. Silguero, Julianne J. Hopper, Abir I. Chowdhury.

**Formal analysis:** Martha M. Werler.

**Investigation:** Charlie B. Fischer, Nedghie Adrien, Jeremiah J. Silguero, Julianne J. Hopper, Abir I. Chowdhury, Martha M. Werler.

**Methodology:** Charlie B. Fischer, Nedghie Adrien, Jeremiah J. Silguero, Julianne J. Hopper, Abir I. Chowdhury, Martha M. Werler.

**Visualization:** Charlie B. Fischer, Jeremiah J. Silguero, Julianne J. Hopper, Abir I. Chowdhury, Martha M. Werler.

**Writing – original draft:** Charlie B. Fischer, Jeremiah J. Silguero, Julianne J. Hopper, Abir I. Chowdhury, Martha M. Werler.

**Writing – review & editing:** Charlie B. Fischer, Nedghie Adrien, Jeremiah J. Silguero, Julianne J. Hopper, Abir I. Chowdhury.

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
