## [Decision Letter · Decision Letter 0]

29 Jan 2021

PONE-D-21-01163

Mask adherence and rate of COVID-19 across the United States

PLOS ONE

Dear Dr. Werler,

Thank you for submitting your manuscript to PLOS ONE. After careful consideration, we feel that it has merit but does not fully meet PLOS ONE’s publication criteria as it currently stands. Therefore, we invite you to submit a revised version of the manuscript that addresses the points raised during the review process.

We look forward to receiving your revised manuscript.

Kind regards,

Wen-Jun Tu

Academic Editor

PLOS ONE

Journal Requirements:

Reviewers' comments:

Reviewer's Responses to Questions

**Comments to the Author**

1. Is the manuscript technically sound, and do the data support the conclusions?

Reviewer #1: Partly

Reviewer #2: Partly

2. Has the statistical analysis been performed appropriately and rigorously? 

Reviewer #1: Yes

Reviewer #2: No

3. Have the authors made all data underlying the findings in their manuscript fully available?

Reviewer #1: Yes

Reviewer #2: No

4. Is the manuscript presented in an intelligible fashion and written in standard English?

Reviewer #1: Yes

Reviewer #2: No

5. Review Comments to the Author

Reviewer #1: This is an interesting paper that adds important evidence to the debate over the value of masks in reducing the spread of COVID-19. The association between self-reported mask-wearing and changes in reported cases are particularly intriguing. However, there are several aspects of the analysis that could benefit from clarification or revision.

As the authors note, using quartiles to measure mask adherence provides relative rather than absolute values for compliance in each state. It would be interesting to know the actual percentages of compliance. Perhaps this could be included in an additional table.

Use of the cut-off value of 15% for the risk factors used as covariates should be explained further. What is the reason it was chosen?

The authors should more clearly acknowledge the large amount of variation within states. Racial and ethnic groups, for example, may be concentrated in urban areas. As a result, the overall percentages may be of limited value as covariates.

Within-state variation may explain the large confidence intervals in the odds ratios in the adjusted models. This calls into question the value of the results. The authors should explain more clearly why this analysis is valuable or consider deleting it.

The authors include race and ethnicity as covariates but not income. Poverty is a clear risk factor for COVID-19 and may account for much of the variation attributed to race and ethnicity. The authors should consider using it in addition to, or instead of, race and ethnicity.

The grouping together of state mask policies (recommendations and mandates) should be discussed further. There is a considerable difference between recommending and mandating a behavior. The authors should consider analyzing each kind of policy separately or focusing only on mandates, which are more important from a policy perspective.

It would be interesting to consider the association between mask policies and adherence. Figures 1 and 2 suggest they have similar effects on case counts, which is not surprising. It is likely that mask policies drive adherence. It is also possible that they both reflect a common factor, such as a state’s political orientation. The authors might consider analyzing, or at least discussing, their interaction.

Some of the limitations that are mentioned are important and should be given another sentence or two of discussion. Self-reporting of mask adherence could be an important source of bias in responses. In states with mandates, respondents would be more likely to want to appear compliant. Reverse causation could also have an important effect. High case counts in a state could induce more people to wear masks.

I also have a few more minor comments.

The references to resistance to mask mandates and the effect of social media in the Introduction should have citations.

The survey that estimated mask adherence should be explained further. How regularly was it conducted, how were respondents selected and how were they contacted?

The source of state population estimates should be stated.

It would be interesting to see the number of states falling into the high and low categories for mask adherence.

There is a typo in the last line before figures 1 and 2 (“none” instead of “no”). There also seems to be a typo in the heading for figure 2 (an extra “with”).

The heading for table 2 is unclear. Does it mean the odds of a high case count from one month to the next?

The reference for odds ratios being “decreased” is confusing. Does it mean they were lower in the adjusted models?

The reference to “increases” in case counts from one month to the next should be replaced with “changes”, since they could go down. The word “minoritized” should be changed to “minority”.

Reviewer #2: see attached, while websites are cited for point 3 above, need to note date that the data was actually obtained so that the exact dataset can be accessed. There appears to be some errors in Table 1 that should be corrected

6. PLOS authors have the option to publish the peer review history of their article (what does this mean?). If published, this will include your full peer review and any attached files.

Reviewer #1: No

Reviewer #2: No

---

## [Author Response · Author response to Decision Letter 0]

18 Feb 2021

Thank you for this very helpful review. We appreciate your recognition of the many strengths of the paper. Below we respond to each reviewer comment or query with italicized text. Our responses also identify where changes to the manuscript appear, accordingly.

The PLOS Data policy requires authors to make all data underlying the findings described in their manuscript fully available without restriction. 

The data are provided as supplementary file "Other"

As the authors note, using quartiles to measure mask adherence provides relative rather than absolute values for compliance in each state. It would be interesting to know the actual percentages of compliance. Perhaps this could be included in an additional table.

Because the distribution of mask adherence changes each month in each state, the number of actual percentage values is prohibitively large (51 states+DC x 6 months x 4 quartiles). We do state the cut-off values for the low and high quartiles for June, July, August, and September in the results section. 

Use of the cut-off value of 15% for the risk factors used as covariates should be explained further. What is the reason it was chosen?

We have added the following to the methods section “For demographic data, we dichotomized population proportions at whole values that approximated the highest quartile of the distributions. Specifically, we created the following categories: >15% non-Hispanic Black, >15% Hispanic, median age >40 years, and population density >200 people per square mile, which corresponded to 74.5%, 78.4%, 82.4%, and 78.4% of the distributions, respectively.”

The authors should more clearly acknowledge the large amount of variation within states. Racial and ethnic groups, for example, may be concentrated in urban areas. As a result, the overall percentages may be of limited value as covariates. It is important to note that state level distributions of demographic factors do account for concentrations or sparsity of populations within a given state. 

We have added the following sentence to our paragraph on the limitations of our study with respect to demographic factors: “It is important to note that state level distributions of demographic factors do account for concentrations or sparsity of populations within a given state.” 

Within-state variation may explain the large confidence intervals in the odds ratios in the adjusted models. This calls into question the value of the results. The authors should explain more clearly why this analysis is valuable or consider deleting it.

The confidence intervals around odds ratios for adherence measures are not disconcertingly wide. Those for mask policy are wide, due to the less powerful bivariate measurement. We believe the results in table 2 are useful to readers because they show how adherence reduces high COVID rates even after policy is adjusted for and how little confounding there is due to demographic data (at least as far as the demographic factors are measured vis-à-vis the previous reviewer comment). We have added this point to our description of adherence odds ratios in the Results section. The estimated decrease in odds of high COVID-19 for every 1% increase in mask adherence adds strong evidence of the more simplified findings as shown in Figure 1. 

The authors include race and ethnicity as covariates but not income. Poverty is a clear risk factor for COVID-19 and may account for much of the variation attributed to race and ethnicity. The authors should consider using it in addition to, or instead of, race and ethnicity.

We agree that income information would be interesting to examine and perhaps more informative, but data on income were not available in the COVID-19 US state policy database. We do appreciate that social/economic/political factors underlie why certain population groups have higher rates of COVID-19, which we were not able to address in our analysis. We discuss this issue in the penultimate paragraph of the Results and Discussion section.

The grouping together of state mask policies (recommendations and mandates) should be discussed further. There is a considerable difference between recommending and mandating a behavior. The authors should consider analyzing each kind of policy separately or focusing only on mandates, which are more important from a policy perspective.

Thank you for this suggestion. We have added to Figure 2 bar graphs for states with strict and with recommended mask wearing policies. Due to small numbers, regression models for mask-wearing policy compare any policy versus no policy. 

It would be interesting to consider the association between mask policies and adherence. Figures 1 and 2 suggest they have similar effects on case counts, which is not surprising. It is likely that mask policies drive adherence. It is also possible that they both reflect a common factor, such as a state’s political orientation. The authors might consider analyzing, or at least discussing, their interaction.

Mask adherence is higher in states with mask policy, but regression models to estimate statistical interaction produced unreliable coefficients. The variability across states provided us the opportunity to measure independent effects of each on odds of high COVID-19 rates as shown in Table 2. We did stratify the data according to policy categories and observed reduced odds ratios associated with mask adherence for states with policy and for states without policy. Even among states with mask policy, associations between increasing mask adherence and high COVID-19 rates were evident. We have added a sentence to the Results section, describing these stratified results.

Some of the limitations that are mentioned are important and should be given another sentence or two of discussion. Self-reporting of mask adherence could be an important source of bias in responses. In states with mandates, respondents would be more likely to want to appear compliant. Reverse causation could also have an important effect. High case counts in a state could induce more people to wear masks.

Regarding reverse causation, we discuss this possibility with respect to our consideration of temporality. We have added a note that reporting error might be influenced by policy. Rather than a cross-sectional design, we looked at mask adherence in the month preceding COVID-19 rates. 

The references to resistance to mask mandates and the effect of social media in the Introduction should have citations. 

We added reference #6. 

The survey that estimated mask adherence should be explained further. How regularly was it conducted, how were respondents selected and how were they contacted? 

A more detailed description is added to the Methods.

The source of state population estimates should be stated. 

This reference was added.

It would be interesting to see the number of states falling into the high and low categories for mask adherence. 

Please see our response to this reviewer’s first query about quartiles of adherence.

There is a typo in the last line before figures 1 and 2 (“none” instead of “no”). There also seems to be a typo in the heading for figure 2 (an extra “with”). 

These have been corrected.

The heading for table 2 is unclear. Does it mean the odds of a high case count from one month to the next?

We changed the title of Table 2 to: State-level odds ratios and 95% confidence intervals (CI) for mask adherence and mask policy in relation to high COVID-19 rates in the subsequent month

The reference for odds ratios being “decreased” is confusing. Does it mean they were lower in the adjusted models? 

We modified this sentence to help the reader interpret odds ratios: “For every 1% increase in average adherence in June, the fully adjusted odds ratios for high COVID-19 in July was 0.95, indicating a protective effect against high COVID-19 rates.“ 

The reference to “increases” in case counts from one month to the next should be replaced with “changes”, since they could go down. 

We modified this sentence to read: “…in a given month does not capture the most effective time period that influences COVID-19 rates.”

The word “minoritized” should be changed to “minority”. 

Minoritize means to make a minority, as distinguished from being a minority. In this context, populations with the greatest risk for COVID-19 might constitute the majority in a given geographic area but are minoritized in a social context. 

Reviewer #2: see attached, while websites are cited for point 3 above, need to note date that the data was actually obtained so that the exact dataset can be accessed. There appears to be some errors in Table 1 that should be corrected. `

These errors have been corrected. 

Queries:

1. Table 1 broadly categorizes the states and DC according to their demographics, mask-wearing policies, mask-wearing adherence, and COVID-19 cases. However, this table inaccurately lists WA and OR as among the states with no mask policy from April-August. WA implemented mandatory masking policies on June 26, 2020, and July 24, 2020, and at least one of these orders met the authors’ criteria as a “strict” policy. Likewise, OR implemented initial mandatory masking policies on July 1, 2020. We have not checked other state categorizations in this table, but these errors suggest the possibility that other errors may exist in the data and that the analyses in the manuscript may be flawed. 

There was indeed an error in Table 1 for the rows that list states with on mask wearing policy. We corrected that error in the table. We also verified that the coding was correct for the data presented in Figure 2 and from models 2 and 3 in Table 2. One correction was made to Figure 2, where there were 11 states (not 10) with high COVID-19 rates among the 15 states with no mask policy in the month of August. We added additional rows to Table 1 to show the states with high COVID-19 rates by month.

2. IHME projects that 95% mask use reduces COVID-19 case rate by 30% or more, whereas the observed rate of mask use in August and September 2020 nationally was approximately 65%. Following IHME’s projections, that suggests an anticipated reduction in COVID-19 rate that would be less than the above estimated 30%. Given that this study analyzed mask use and COVID-19 rate for each state, it would be useful if the authors provided and clearly stated COVID-19 rate reductions, even though they may vary by state. 

We agree that it would be interesting to use these data to make projections like IHME, but we did not calculate moving averages or multilevel simulations. We have added to the abstract and results the mean COVID-19 rates for states with >75% masking in a given month versus the others. “Mean COVID-19 rates for states with at least 75% mask adherence in the preceding month was 109.26 per 100,000 compared to 249.99 per 100,000 for those with less adherence.”

3. The authors need to carefully compare the description of their results to their methods and to the depiction of their results in Figures 1 and 2, as they do not seem to match. 

First, in the text they state that 16, 18, 16, and 30 states had high rates of adherence in the months of June, July, August, and September, respectively, whereas Figure 1 seems to show that only 13 states had high rates of adherence for these months. It seems likely that the text description reflects an inconsistent or confusing application of their quartile method of categorizing mask adherence—indeed, isn’t it nonsensical to describe 16, 18, or 30 states as having high adherence given their previous description of high adherence as belonging to the upper quartile, which would presumably consist of 13 states? It may be useful for the authors to either reconsider this quartile method or to offer more insight into this analytic method elsewhere in the paper and to check that it is consistently applied to all mentions of mask-wearing adherence. 

Thank you for identifying areas where our descriptions of methods and results were confusing. In fact the description 16, 18, 16, and 30 states with high rates referred to high COVID-19 rates, but we didn’t explicitly state this. That sentence now reads: “Across these four months, the proportion of states with COVID rates in the high category were 19 (37%), 19 (37%), 20 (39%), and 32 (63%), respectively. “ 

Second, the sentence about Arizona, Connecticut, Hawaii, Massachusetts, Michigan, New York, Rhode Island, Vermont, and DC is unclear; it should presumably begin by stating that 9 of the 13 states/DC in the high-adherence quadrant did not experience a high COVID-19 rate in the subsequent month. 

We appreciate that our statements were confusing. Because the cut-off for each quartile differed by month, we added a separate analysis that was anchored to a set level (>75%) of adherence. We have added this analytic step to the methods section. We have also changed the labeling for ‘high’ adherence to ‘highest quartile’ to emphasize that the inter-quartile ranges vary from month to month. 

Third, the text related to Figure 2 speaks of “no” mask wearing policy and “any” mask wearing policy, whereas Figure 2 itself depicts “no” mask wearing policy and “some” mask wearing policy. Given the description in the Methods section, the Figure 2 language of “some” presumably would include both “recommended” and “strict” mask wearing policies, though this is certainly not spelled out. Moreover, the percentages mentioned in the text, do not necessarily seem to match the percentages depicted in the figure. For instance, the text indicates that 40 to 73% of no policy states had high infection rates, whereas the figure visually seems to show 50 to 73%. Likewise, the text indicates that less than 20% of states with a mask-wearing policy had high COVID-19 rates in the first three months studied, whereas the figure seems to depict about 25% for these states for July/August and August/September.

Thank you for noting this inconsistency in our labeling of mask policies. We have changed the labels in Figure 2 to correspond with the text by replacing ‘some’ with ‘any. In addition, we have made the two corrections noted in the text. 

4. It may be useful for the authors to comment in the discussion section on the notable spike seen in October for COVID-19 infections among states that had some mask-wearing policies in place in September. That is, do the authors draw any implications from the data or from other factors for the potential decrease in the efficacy of masking behaviors/policy compared to non-masking behaviors/policy for that month?

This is an excellent point. For any given month, states with >75% mask adherence did not have COVID-19 rates in the high category in the subsequent month. We have added to Table 1 the list of states with >75% adherence and COVID-19 rates <200/100,000 in the subsequent month. Interestingly, mask adherence decreased for many states from August to September and there were no states with that benchmark (>75%) in September. We have added this observation “It is worth noting that no states achieved this level of mask adherence in September, which might account in part for the spike in COVID-19 rates in October. “

---

## [Decision Letter · Decision Letter 1]

4 Mar 2021

PONE-D-21-01163R1

Mask adherence and rate of COVID-19 across the United States

PLOS ONE

Dear Dr. Werler,

Thank you for submitting your manuscript to PLOS ONE. After careful consideration, we feel that it has merit but does not fully meet PLOS ONE’s publication criteria as it currently stands. Therefore, we invite you to submit a revised version of the manuscript that addresses the points raised during the review process.

We look forward to receiving your revised manuscript.

Kind regards,

Wen-Jun Tu

Academic Editor

PLOS ONE

Journal Requirements:

Additional Editor Comments (if provided):

1. In order to provide a more complete information to our readers on the topic, we would like to emphasize the importance to cross referencing very recent material on the same topic published in "PLoS ONE ". Therefore, it would be highly appreciated if you would check the contents published in the last two years of "PLoS ONE" (https://journals.plos.org/plosone/) and add all material relevant to your article to the reference list.

2. Add “Clinical Features and Short-term Outcomes of 102 Patients with Corona Virus Disease 2019 in Wuhan, China. Clinical Infectious Diseases, 71(15):748-755‘ in revision text

Reviewers' comments:

Reviewer's Responses to Questions

**Comments to the Author**

1. If the authors have adequately addressed your comments raised in a previous round of review and you feel that this manuscript is now acceptable for publication, you may indicate that here to bypass the “Comments to the Author” section, enter your conflict of interest statement in the “Confidential to Editor” section, and submit your "Accept" recommendation.

Reviewer #1: (No Response)

Reviewer #2: All comments have been addressed

2. Is the manuscript technically sound, and do the data support the conclusions?

Reviewer #1: Yes

Reviewer #2: Yes

3. Has the statistical analysis been performed appropriately and rigorously? 

Reviewer #1: Yes

Reviewer #2: Yes

4. Have the authors made all data underlying the findings in their manuscript fully available?

Reviewer #1: Yes

Reviewer #2: Yes

5. Is the manuscript presented in an intelligible fashion and written in standard English?

Reviewer #1: Yes

Reviewer #2: Yes

6. Review Comments to the Author

Reviewer #1: The article is greatly improved. However, clarification of a few points would still be helpful.

The phrase “government leaders” in the first paragraph is vague. It could apply to a wide range of officials from governors to city health commissioners, who have different roles and levels of influence. It would better to simply say “…more and more states encourage…”

I found table 1 difficult to follow. Some suggestions –

• The title would be clearer if it read “States with high COVID-19 population risk characteristics.”

• The headings “Not high COVID-19 rate in subsequent month” would be clearer if they read “States without high COVID-19 rate in month subsequent to high mask adherence.” As written, it is not immediately apparent what the month is subsequent to.

• An extra blank row could be added between each heading to make it easier for the reader to follow.

• The table could be broken into two – one for rates >200 cases/100,000 and one for rates >50 cases/100,000. As presently structured, the distinction is buried in footnotes.

The title for table 2 would be clearer if the words “versus lower” were deleted. The comparison of higher rates to lower rates is implied.

As a suggestion, did the authors consider conducting an analysis with lower rates rather than high rates as the outcome? It might make the point more clearly that mask adherence is associated with lower rates.

A slight wording change in the last sentence under Results would make it clearer. The word “of” should be replaced with “against” to read “… protection against high…”

The first paragraph under Interpretation seems to be saying that issuing a policy is not the same as successfully implementing it. The paragraph would be clearer this were stated directly.

The last sentence in the third paragraph under Interpretation would be clearer if it read “…should reduce the chance of reverse causation…”

The sentence on demographic factors in the fourth paragraph under Interpretation would be clearer if it simply stated that the demographic factors considered in the analysis may be surrogates for socioeconomic disadvantages.

The last two sentences in that paragraph, which refer to access to testing, should acknowledge another possible limitation of the study. Variation in case counts might reflect variation in the extent of testing.

Reviewer #2: (No Response)

7. PLOS authors have the option to publish the peer review history of their article (what does this mean?). If published, this will include your full peer review and any attached files.

Reviewer #1: No

Reviewer #2: No

---

## [Author Response · Author response to Decision Letter 1]

15 Mar 2021

Reviewer #1: The article is greatly improved. However, clarification of a few points would still be helpful.

The phrase “government leaders” in the first paragraph is vague. It could apply to a wide range of officials from governors to city health commissioners, who have different roles and levels of influence. It would better to simply say “…more and more states encourage…”

We have changed that sentence as suggested in both the abstract and introduction.

I found table 1 difficult to follow. Some suggestions –

• The title would be clearer if it read “States with high COVID-19 population risk characteristics.”

We have changed that title as suggested.

• The headings “Not high COVID-19 rate in subsequent month” would be clearer if they read “States without high COVID-19 rate in month subsequent to high mask adherence.” As written, it is not immediately apparent what the month is subsequent to.

We have changed that row label to: “COVID-19 rate <200 cases/100,000 in month subsequent to high mask adherence.”

• An extra blank row could be added between each heading to make it easier for the reader to follow.

We now separate each row with a bottom border line.

• The table could be broken into two – one for rates >200 cases/100,000 and one for rates >50 cases/100,000. As presently structured, the distinction is buried in footnotes.

We have separated table 1 into two tables. New table 1 is on risk characteristics. New table 2 is entitled “States with high COVID-19 rates.” We added the following sentence to introduce the new table 2: “The list of states with high COVID-19 rates by month shows the initial wave in northeastern states in May, followed by a wave in southern states, and then spreading across the U.S. over the next four months (Table 2).” We removed the last row of the table and now describe those data in the text: “Eight states were reported to have at least 75% mask adherence in any month between June and September (AZ, CT, HI, MA, NY, RI, VT, VA); none reported a high COVID-19 rate in the subsequent month.” 

The title for table 2 would be clearer if the words “versus lower” were deleted. The comparison of higher rates to lower rates is implied.

[This is now Table 3 due to the new Table 2.] We deleted ‘versus lower’ from its title. 

As a suggestion, did the authors consider conducting an analysis with lower rates rather than high rates as the outcome? It might make the point more clearly that mask adherence is associated with lower rates.

We elected to evaluate high mask adherence, rather than low adherence, to align with the public health recommendation. In addition to the associations we observed, we were able to identify a level for states to target (>75% adherence) where lower COVID-19 rates followed. 

A slight wording change in the last sentence under Results would make it clearer. The word “of” should be replaced with “against” to read “… protection against high…”

We have made this change.

The first paragraph under Interpretation seems to be saying that issuing a policy is not the same as successfully implementing it. The paragraph would be clearer this were stated directly.

We added the following sentence to that paragraph: “Indeed, issuing such a policy is not the same as successfully implementing it.”

The last sentence in the third paragraph under Interpretation would be clearer if it read “…should reduce the chance of reverse causation…”

We have modified that sentence accordingly.

The sentence on demographic factors in the fourth paragraph under Interpretation would be clearer if it simply stated that the demographic factors considered in the analysis may be surrogates for socioeconomic disadvantages.

Given the potential for implicit or explicit bias that can result from how race and ethnicity are conceptualized, operationalized, and interpreted in statistical analyses, we believe it is important to provide the reader with this more detailed discussion. 

The last two sentences in that paragraph, which refer to access to testing, should acknowledge another possible limitation of the study. Variation in case counts might reflect variation in the extent of testing.

We have modified the last sentence to read: “Our study was also limited by the lack of information on accessibility of COVID-19 testing; if less accessible testing is associated with less mask adherence, the associations we report here may be under-estimates.”

1. We note that you currently have two Tables in your manuscript titled as Table 2.

The first "Table 2. States with high COVID-19 rates." - and the second "Table 2: State-level odds ratios and 95% confidence intervals (CI) for high versus lower COVID-19 rates in the subsequent month".

* So that these tables can be differentiated can you please update the Table title numbering and the in-text citations to them accordingly.

---

## [Editor Report · Decision Letter 2]

18 Mar 2021

PONE-D-21-01163R2

Mask adherence and rate of COVID-19 across the United States

PLOS ONE

Dear Dr. Werler,

Thank you for submitting your manuscript to PLOS ONE. After careful consideration, we feel that it has merit but does not fully meet PLOS ONE’s publication criteria as it currently stands. Therefore, we invite you to submit a revised version of the manuscript that addresses the points raised during the review process.

We look forward to receiving your revised manuscript.

Kind regards,

Wen-Jun Tu

Academic Editor

PLOS ONE

Journal Requirements:

Additional Editor Comments (if provided):

1. In order to provide a more complete information to our readers on the topic, we would like to emphasize the importance to cross referencing very recent material on the same topic published in "PLoS ONE ". Therefore, it would be highly appreciated if you would check the contents published in the last two years of "PLoS ONE" (https://journals.plos.org/plosone/) and add all material relevant to your article to the reference list.

2. Add “Clinical Features and Short-term Outcomes of 102 Patients with Corona Virus Disease 2019 in Wuhan, China. Clinical Infectious Diseases, 71(15):748-755‘ in revision text

---

## [Author Response · Author response to Decision Letter 2]

18 Mar 2021

We have added your paper to the citation list and a publication in PLoS One on perceptions of mask wearing. We also noted that one citation was incomplete and have corrected that. We checked all other references and none have been retracted.

---

## [Editor Report · Decision Letter 3]

29 Mar 2021

Mask adherence and rate of COVID-19 across the United States

PONE-D-21-01163R3

Dear Dr. Werler,

We’re pleased to inform you that your manuscript has been judged scientifically suitable for publication and will be formally accepted for publication once it meets all outstanding technical requirements.

Kind regards,

Wen-Jun Tu

Academic Editor

PLOS ONE
---

## [Editor Report · Acceptance letter]

1 Apr 2021

PONE-D-21-01163R3 

Mask adherence and rate of COVID-19 across the United States 

Dear Dr. Werler:

I'm pleased to inform you that your manuscript has been deemed suitable for publication in PLOS ONE. Congratulations! Your manuscript is now with our production department. 

Kind regards, 

on behalf of

Dr. Wen-Jun Tu 

Academic Editor

PLOS ONE